# Terpenoids of the Swamp Cypress Subfamily (Taxodioideae), Cupressaceae, an Overview by GC-MS

**DOI:** 10.3390/molecules24173036

**Published:** 2019-08-21

**Authors:** Bernd R. T. Simoneit, Angelika Otto, Daniel R. Oros, Norihisa Kusumoto

**Affiliations:** 1Department of Chemistry, College of Science, Oregon State University, Corvallis, OR 97331, USA; 2Forschungsinstitut Senckenberg, Sektion Paläobotanik, D-60325 Frankfurt am Main, Germany; 3Consultant, 72 Marina Lakes Drive, Richmond, CA 94804, USA; 4Wood Extractive Laboratory, Forestry and Forest Products Research Institute, Tsukuba, Ibaraki 305-8687, Japan

**Keywords:** *Cryptomeria japonica*, *Glyptostrobus pensilis*, *Taxodium distichum*, *Taxodium mucronatum*, Cupressaceae, GC-MS overview, terpenoids

## Abstract

The resins bled from stems and in seed cones and leaves of *Cryptomeria japonica, Glyptostrobus pensilis*, *Taxodium distichum*, and *T. mucronatum* were characterized to provide an overview of their major natural product compositions. The total solvent extract solutions were analyzed as the free and derivatized products by gas chromatography-mass spectrometry to identify the compounds, which comprised minor mono- and sesquiterpenoids, and dominant di- and triterpenoids, plus aliphatic lipids (e.g., *n*-nonacosan-10-ol). Ferruginol, 7α-p-cymenylferruginol, and chamaecydin were the major characteristic markers for the Taxodioideae conifer subfamily. The mass spectrometric data can aid polar compound elucidation in environmental, geological, archeological, forensic and pharmaceutical studies.

## 1. Introduction

Natural products, especially terpenoids or their derivatives, are preserved in the ambient environment or geological record. When extracted and characterized by gas chromatography-mass spectrometry (GC-MS) they are used by organic geochemists as tracers for sources, transport and alteration processes of organic matter in any global compartment [1,2,3,4,5,6,7,8,9,10,11,12,13,14,15]. The application of GC-MS in the analysis of natural product mixtures extracted from plants for compound characterization can also be of utility for rapid screening in pharmacological studies [16]. 

The Coniferae are known as important source plants for resins and are comprised of the Araucariaceae (3 genera), Cupressaceae (27 genera), Pinaceae (11 genera), Podocarpaceae (18 genera), Taxaceae (6 genera), and Sciadopityaceae (1 genus) [17]. Here we focus on the Cupressaceae, specifically the subfamily Taxodioideae with 4 species (Table 1). Fossil remains of *Taxodium* and *Glyptostrobus* leaflets, seed cones and wood are often found in the geologic record of the northern hemisphere since the Cretaceous, and are especially abundant in the paleofloras of Tertiary brown coals [18,19,20,21,22] and extant peats of Florida, USA [23].

Structure determinations and pharmacological potential studies of resin terpenoids isolated and purified from species of the Taxodioideae subfamily have been reported by numerous chemists and pharmacologists [24,25,26,27,28,29,30,31,32,33,34,35,36,37,38,39,40,41,42]. However, the connection of the known natural products isolated from these conifers with their application as tracers in other interdisciplinary sciences has been limited [7,8,14,43]. This is due to the lack of reports on the chemical compositions of the total resin mixtures, and the paucity of mass spectra for known compounds, especially the oxygenated natural products. Here we characterize the total compound mixtures in resins from the Taxodioideae subfamily based on correlation and comparison with known standards. We report the GC-MS results and full mass spectra of novel compounds of the resins as free and derivatized products. This total mixture analysis provides an overview of the major compounds present in the resins of the extant Taxodioideae subfamily, and is a guide for their presence in environmental and fossil samples.

## 2. Results and Discussion

The contents of terpenoid natural products have been documented in many species of the Cupressaceae [17,44,45,46], but the investigation of the former family Taxodiaceae is incomplete. The terpenoid content of *Glyptostrobus pensilis* (Staunton) Koch (Chinese Water Pine) is limited [17], with only two trace diterpenoids, namely glypensin A and 12-acetoxy-*ent*-labda-8(17),13E-dien-15-oic acid, identified [37]. According to the merger of Cupressaceae and Taxodiaceae to the Cupressaceae s.l., *G. pensilis* was placed in the subfamily Taxodioideae [47].

The diterpenoid constituents of *Cryptomeria japonica* and *Taxodium distichum* have been studied more extensively [24,25,26,27,28,29,31,32,33,34,35,38,39,40,41,42,48] and references therein]. However, none of these earlier studies reported the natural product compositions of the total resin/plant extracts, but instead the mixtures were separated by liquid or high-pressure liquid chromatography (LC or HPLC), followed by structure determination of each compound by NMR, HRMS for elemental composition, and sometimes MS (underivatized by direct insertion probe). The diterpenoid components of *Taxodium mucronatum* have had limited examination [7,8,49].

The compounds identified as significant components in the bled resins (as total extracts) of the four conifers in this subfamily (Table 1) are listed here in Table 2. Analyses of extracts of leaves and cones yielded similar results, except the solvent mixture used for extraction also isolated the epicuticular wax and polar cell (e.g., saccharides) components which were superimposed on the terpenoids. Examples of annotated total ion current traces for the GC-MS results are shown in Figure 1. The mass spectra of the major compounds in the extract mixtures are given in the figures of the Appendix A section or can be found in the literature cited in Table 2.

### 2.1. Cryptomeria

The resin of *Cryptomeria japonica* (sugi cedar) was comprised of minor diterpenoids as the phenolic abietane-type: Ferruginol (*a2*, all chemical structures are given in Appendix B), 6,7-dehydroferruginol (*a1*), sugiol (*a4*), 7α-p-cymenylferruginol (*a26*), chamaecydin (*a27*), *iso*-chamaecydin (*a28*), and 6-hydroxychamaecydin (*a29*); and major diterpenoid resin acids with the pimarane and labdane skeletons: Sandaracopimaric acid (*p1*), isopimaric acid (*p2*, 100%), *iso*-communic acid (*L3*), 13-*epi*-cupressic acid (*L5*), imbricataloic acid (*L4*), imbricatoloic acid (*L7*), *iso*-cupressic acid (*L6*), 13,14-dihydroagathic acid (*L8*), and 13E- and 13Z-communic acids (*L2*). No sesquiterpenoids were detectable, and the lipid compound, *n*-nonacosan-10-ol (*n1*), was a carryover from the epicuticular leaf wax.

The essential oil of *C. japonica* has been analyzed, but only mono- and sesquiterpenoids were reported [50]. Kaurenes and dehydroabietane with the lower terpenoids were reported for sugi leaf oil [51]. The minor resin acids have been characterized in bark and leaf extracts [27,28,29], and the chamaecydins were elucidated in leaf extracts from *C. japonica* [25,26]. The diterpenoids from bark and fruits of *C. fortunei*, also known as *C. japonica* var. *sinensis*, have been studied [36,52]. Sugiol (*a4*), 11-hydroxysugiol (*a15*), and 14-deoxycoleon U (*a17*), also found in this study, were reported.

### 2.2. Glyptostrobus

The resin of *Glyptostrobus pensilis* consisted mainly of diterpenoids as the phenolic abietanes: Ferruginol (*a2*), 6,7-dehydropisiferol (*a3*), pisiferol (*a7*), *abeo*-pisiferol (*a8*), *abeo*-carnosol (*a16*, 100%), sugiol (*a4*), salvinolone (*a11*), and 6-hydroxysalvinolone (14-deoxycoleon U, *a17*) as major compounds. The minor compounds were 6,7-dehydroferruginol (*a1*), 7-hydroxytaxodone (*a19*), sandaracopimaric acid (*p1*), 7α-p-cymenylferruginol (*a26*), chamaecydin (*a27*), *iso*-chamaecydin (*a28*), 6-hydroxychamaecydin (*a29*), and 19(20)-oxoferruginol (*a5*), also mostly phenolic abietanes. The epicuticular wax extracted from shoots of *G. pensilis* was comprised of lipids, mainly *n*-nonacosan-10-ol (*n1*), *n*-nonacosan-10-one (*n2*) and *n*-alkanes ranging from C_27_ to C_33_; with minor diterpenoids: Ferruginol (*a2*, 100%), 6,7-dehydroferruginol (*a1*), 6,7-dehydropisiferol (*a3*), pisiferol (*a7*), sugiol (*a4*), salvinolone (*a11*), 6-hydroxysalvinolone (*a17*), *abeo*-carnosol (*a16*), *abeo*-pisiferol (*a8*), and triterpenoids: 24-methylenecycloartan-3-one (*T1*), and 24-ethylenecycloartan-3-one (*T2*). No sesquiterpenoids were detected.

The terpenoids of oil steam distilled from wood of *G. pensilis* from Vietnam were comprised of mono- and sesquiterpenoids with traces of phytol and ferruginol [63]. However, fossil leaves of this conifer back to the Eocene retained mainly the following natural products: ferruginol, 6,7-dehydroferruginol, salvinolone, 5,6-dehydrosugiol, taxodione acetate, *abeo*-carnosol, sugiol, cymenylferruginol, chamaecydin, and iso-chamaecydin [7,8,54,64,65].

### 2.3. Taxodium distichum

The resin of *Taxodium distichum* was comprised mainly of phenolic abietanes: Ferruginol (*a2*), 6,7-dehydroferruginol (*a1*), sugiol (*a4*), taxoquinone (*a18*), taxodione (*a10*, R=H), 6α-hydroxytaxoquinone (*a23*), 6β-hydroxytaxoquinone (*a24*), taxodone (*a14*, 100%), 11-hydroxysugiol (*a15*), 7-hydroxytaxodone (*a19*), 6-hydroxysalvinolone (*a17*), salvinolone (*a11*), and chamaecydin (*a27*). The other minor compounds that could be identified were: 6-deoxotaxodione (*a6*), 7α-p-cymenylferruginol (*a26*), sandaracopimaric acid (*p1*), *iso*- and 6-hydroxychamaecydin (*a28, a29*), royleanone (*a13*), taxodione acetate (*a10*, R=Ac), and 7-acetoxy-6,7-dehydroroyleanone (*a25*). 

These compounds, with numerous trace components, have all been reported in the extensive literature [24,31,32,33,34,38,66], also as cited in Table 2. *Taxodium* peat has been analyzed but only aromatic hydrocarbons derived from diterpenoids were reported [23]. Discrete fossils of *Taxodium* back to the Eocene contained ferruginol, 6,7-dehydroferruginol, taxodione acetate, sugiol, and the chamaecydins [7,8,65,67,68].

### 2.4. Taxodium mucronatum

The *T. mucronatum* resin consisted of significant sesquiterpenoids: Widdrol (*s4*), widdrene (*s2*), mannol (*L1*), cuparene (*s1*), and mayurone (*s3*). The diterpenoids were exclusively only phenolics of the abietane-type: Ferruginol (*a2*, 100%), 6,7-dehydroferruginol (*a1*), 6-deoxotaxodione (*a6*), 7-acetoxyroyleanone (*a25*), taxodone (*a14*), taxodione acetate (*a10*, R = Ac), and chamaecydin (*a27*). The minor components were: Royleanone (*a13*), sugiol (*a4*), salvinolone (*a11*), 7α-p-cymenylferruginol (*a26*), *iso*- and 6-hydroxychamaecydin (*a28, a29*), and the lipid *n*-nonacosan-10-ol (*n1*). One study reported the presence of 8β-hydroxypimar-15-en-19-oic acid in this plant [49].

### 2.5. Mass Spectrometry

The mass spectra of all compounds analyzed by GC-MS are listed in Table 2. Those indicated by S match with the respective reference standards, and those indicated by L match with the literature data cited.

The mass spectra of 6,7-dehydroferruginol (*a1*), as the free compound and methyl ether [12,65,69,70], as well as the NMR data [27,71] have been published. The interpretation of the mass spectrometric fragmentation, specifically the even mass fragment ion from the retro-Diels–Alder rearrangement loss of ring-A [56] is unusual and should indicate 2,3-dehydroferruginol. The same loss of ring-A by retro-Diels–Alder rearrangement of the C-2 to C-3 double bond is observed in mass spectra of standard cholest-2-ene (loss of C_4_H_6_ from M ^+^ 370 to *m/z* 316), friedel-2-ene (loss of C_5_H_8_ from M ^+^ 410 to *m/z* 342), or lupa-2,22(29)-diene (loss of C_6_H_10_ from M ^+^ 408 to *m/z* 326). This would be consistent for 2,3-dehydroferruginol (loss of C_6_H_10_ from M ^+^ 284 to *m/z* 202, Figure 2a) and its silyl derivative (loss of C_6_H_10_ from M ^+^ 356 to *m/z* 274, Figure 2d). However, the NMR data indicate the double bond is at C-6 to C-7 [71], so that bond can also induce fragmentation by H-transfer and loss of ring-A to *m/z* 202 (Figure 3a). Therefore, the double bond position could be misinterpreted based solely on mass spectra, but with the NMR data the assignment as 6,7-dehydoferruginol is correct.

In fossil resins of these conifers we also found 5,6-dehydroferruginol (tentative identification [7,8,67]), where the elimination of C_4_H_8_ to *m/z* 228 with secondary loss of C_2_H_4_ to *m/z* 200 (or direct loss of ring-A to *m/z* 200) are evident for the free compound and the same fragmentation for the silyl derivative to *m/z* 300 (Figure 2b,e and Figure 3a). Another isomer was tentatively identified as 15,16-dehydroferruginol in the same fossil resins, but not reported before (Figure 2c). Its key fragment ion is *m/z* 215 from loss of C_5_H_9_ in ring-A, as confirmed in the mass spectrum of its silyl derivative where the same loss of C_5_H_9_ results in the fragment ion at *m/z* 287 (Figure 2f and Figure 3a). We find 6,7-dehydroferruginol and ferruginol primarily in extant conifers, and the tentatively assigned 5,6-dehydroferruginol and 15,16-dehydroferruginol with ferruginol and minor 6,7-dehydroferruginol in fossil conifer resins.

The mass spectra of 7α-p-cymenylferruginol (i.e., 7α-p-isopropyl-benzyl-ferruginol, *a26*) and its trimethylsilyl derivative (Figure 4d,f) were identified based on the structure determination by NMR and HRMS [36]. The compound has a low intensity molecular ion (M ^+^ at *m/z* 418 and loses the p-cymenyl fragment to the base peak at *m/z* 285, which is also reflected in the C_10_H_13_ ion at *m/z* 133 (Figure 3c). Its TMS derivative also has a low intensity M ^+^ at *m/z* 490, with losses of CH_3_ to *m/z* 475 and C_10_H_13_ to the base peak at *m/z* 357, and the corresponding key ion at *m/z* 133 (Figure 3c). We also find a lesser amount (typically 20% of the 7α-isomer) of 7β-p-cymenylferruginol (Figure 4c) in all samples analyzed. It is reported as the total of both isomers in Table 2. The extant sample of *T. mucronatum* and the fossil samples of *T. dubium* and *G. oregonensis* had mass spectra of a minor compound, interpreted as 7α-thymylferruginol with M ^+^ at *m/z* 434 and base peak at *m/z* 285, and for its TMS derivative with M ^+^ at *m/z* 578 and base peak at *m/z* 357 (Figure 4b,e,h, respectively). It has not yet been reported in the natural products literature.

The identification of cymenylferruginol (*a26*), a triterpenoid like the chamaecydins, in both the extant and fossil resins is of interest because natural product chemists are identifying numerous dimer terpenoids, i.e., mono-to-diterpenoid, sesqui-to-diterpenoid and diterpenoid dimers, in extant plants [25,26,36,41,48]. Therefore, the formation mechanism is for example ferruginol reacting at C-7 with cymene to produce 7-cymenylferruginol, 6-deoxotaxodione reacting at C-7 and C-14 with sabinene or thujene to form chamaecydin, cadinols reacting with ferruginol at C-7 to yield the various sesquaterpenoids, or peroxidation of ferruginol to dimers [26,41,48,72,73]. Some of these natural products, if adequately concentrated in the extant biomass, may become preserved as tracers in the geological record, as for example cymenylferruginol reported here.

The resin of *T. distichum* has a significant compound identified in the literature as 11-hydroxy-12-oxoabieta-7,9(11),13-triene (a.k.a. 6-deoxotaxodione, *a6*). Its mass spectrum is shown in Appendix A and matches that reported [55]. The interpretation of the fragmentation pattern is based on the analogous fragmentation of 6,7-dehydroferruginol discussed above and correlation with the GC retention index. It also eliminates C_6_H_10_ from the M ^+^ at *m/z* 300 to *m/z* 218 (Appendix A), indicating that the structure may be 11-hydroxy-6,7-dehydroferruginol (*a6*) or its isomer 11-hydroxy-12-oxoabieta-7,9(11),13-triene (*a6*). Silylation of the *T. distichum* resin extract did not produce a compound with M ^+^ at *m/z* 372, but a mass spectrum of a TMS derivative with M ^+^ at *m/z* 444 that eliminates C_6_H_10_ to *m/z* 362, fitting for 11-hydroxy-6,7-dehydroferruginol-diTMS (Appendix A). The GC-MS results for the standard of 11-hydroxy-12-oxoabieta-7,9(11),13-triene revealed that it had oxidized to primarily taxodione and numerous other polyoxygenated products. However, silylation of this standard did yield a trace derivative with a mass spectrum showing M ^+^ at *m/z* 372 with fragments at *m/z* 357, 329 and 287, but no ion for elimination of C_6_H_10_ at *m/z* 290 (Appendix A). Its structure could be that of 11-trimethylsiloxy-12-oxoabieta-7,9(11),13-triene. Therefore, we conclude that 11-hydroxy-12-oxoabieta-7,9(11),13-triene may isomerize to 11-hydroxy-6,7-dehydroferruginol during the silylation derivatization reaction. Furthermore, the hydroxy functionality at C-11 is preferred rather than at C-14 due to the concurrent presence of 11-hydroxyferruginol (*a9*).

Another compound that occurred as a minor component in extant and fossil *Glyptostrobus* sp. had mass spectra that were misinterpreted as hinokione [54,65]. Based on prior mass spectrometric data for hinokione [74,75] we now tentatively suggest that compound to be 19(20)-oxoferruginol (*a5*). Its mass spectrum has M ^+^ at *m/z* 300 for C_20_H_28_O_2_ with the base peak at *m/z* 189 due to loss of C_7_H_11_O, i.e., rearrangement loss of C-20 and ring-A (Figure 2g and Figure 3b). The same loss is observed in the mass spectrum of its TMS derivative from M ^+^ at *m/z* 372 to the base peak at *m/z* 261 (Figure 2h and Figure 3b). The mass spectrometric data do not match with that of the known formosanoxide (i.e., 7(20)-oxoferruginol [76]). Therefore, 19(20)-oxoferruginol may be a natural product derived from epoxidation to C-19 of pisiferol (*a7*) also present in extant *G. pensilis*.

### 2.6. Key Molecular Tracers for the Taxodioideae

Ferruginol, 6,7-dehydroferruginol, sugiol, 7α-p-cymenylferruginol, chamaecydin, *iso*-chamaecydin, and 6β-hydroxychamaecydin were present as generally significant components in the resins of all four species (Table 2). These natural products are stable and have been reported in environmental and fossil samples [7,8,11,14,54,64,65,77].

Resins from *G. pensilis* contained the following unique natural products in addition to those above: pisiferol, *abeo*-pisiferol, salvinolone, *abeo*-carnosol, 6-hydroxysalvinolone, 19(20)-oxoferruginol, and 6α- and 6β,11-dihydroxysugiols. Resins from Eocene fossils of *G. pensilis* consisted of ferruginol, 6,7-dehydroferruginol, 19(20)-oxoferruginol, sugiol, *abeo*-carnosol, salvinolone, and chamaecydins, but no pisiferol or *abeo*-pisiferol [54]. Therefore, the geological fate of the *abeo*-diterpenoids is not fully known. However, a derivative biomarker, i.e., 10α- and 10β-dehydroicetexane, has been elucidated in sediments and petroleum [78], and it could be derived from pisiferol/*abeo*-pisiferol by reductive dehydroxylation during diagenesis and preservation of organic matter. The resin of extant *G. pensilis* contained traces of 7α-p-cymenylferruginol [36]. It was reported first as an unknown compound in fossil *G. oregonensis* from the Miocene of Emerald Creek, ID (labeled U7 in [65]), then in fossil *G. nordenskioeldi* from the Eocene of Axel Heiberg Island, Nunavut, Canada, but misidentified as roylean-20-oic acid [54]. It is now correctly identified as 7α-p-cymenylferruginol (*a26*).

The resins of the two *Taxodium* species had similar compositions comprised of the compounds present in all four species with additional unique quinoid diterpenoids, specifically: taxodione, royleanone, taxodone, taxoquinone, horminone, 6α- and 6β-hydroxytaxoquinones, taxodione acetate, and 7-acetoxy-6,7-dehydroroyleanone [24]. The natural triterpenoid, 7α-p-cymenylferruginol, was a significant component in the resins of both species (e.g., Figure 3c). Resin from a Miocene *T. dubium* contained ferruginol, 6,7-dehydroferruginol, taxodione acetate, 7-acetoxy-6,7-dehydroroyleanone, sugiol, and 7α-p-cymenylferruginol [65]. Therefore, the phenolic and quinoid diterpenoids are of utility as environmental and geological tracers for source origin.

## 3. Samples and Experimental Methods

### 3.1. Plant Material

The bled resins from trunks and branchlets or seed cones with bled resin blebs of *Cryptomeria japonica, Glyptostrobus pensilis*, *Taxodium distichum,* and *T. mucronatum* were sampled from mature trees with botanical labels in various park and garden reserves (Table 1). The resins were placed in glass vials and cones and branchlets in paper envelopes to allow air drying. We are aware that the structure determinations by natural product chemists typically concentrated on internal compounds extracted from bulk trunk wood, bark, leaves or cones to elucidate the different compound compositions and trace components. However, we examined the external bled compounds in order to determine those that enter the environment before or after tree senescence and may become part of the geological fossil record.

### 3.2. Extraction

Cones and branchlets were cut into pieces and extracted by ultra-sonication for 10 min with dichloromethane:methanol (DCM:MeOH, 1:1 *v*/*v*) and soaking in the solvent mixture for 24 h. The resin samples were dissolved (100% soluble) in the same solvent mixture. The total extracts were filtered through glass fiber filters, and concentrated by use of a rotary evaporator and blow-down with dry nitrogen gas.

### 3.3. Derivatization

Aliquots of total extracts were converted to trimethylsilyl derivatives by reaction with N,O-bis-(trimethylsilyl)trifluoroacetamide (BSTFA, Sigma-Aldrich, St. Louis, MO, USA) and a trace of pyridine for 3 h at 70 °C. The samples were blown down to dryness with nitrogen gas and dissolved in hexane prior to analysis. Other aliquots were treated with trimethylsilyldiazomethane (2M in hexane, Sigma-Aldrich) at room temperature for 30 min to convert carboxylic acids to methyl esters. The excess reagent was reacted with concentrated acetic acid, followed by blow-down with nitrogen gas, and solution in hexane prior to analysis.

### 3.4. GC-MS Analysis

The GC-MS analyses of the total and derivatized extracts were performed on a Hewlett-Packard model 6890 GC coupled to a Hewlett-Packard model 5973 MSD (Palo Alto, CA, USA). Separation was achieved on a DB5 (Agilent, Santa Clara, CA, USA) capillary column (30 or 60 m × 0.25 mm i.d., 0.25 µm film thickness). The GC operating conditions were as follows: temperature hold at 65 °C for 2 min, increase from 65 to 300 °C at a rate of 6 °C min^−1^, with a final isothermal hold for 20 min. Helium was used as carrier gas. The samples were injected splitless (typically 1 µL) with the injector temperature at 280 °C. The MS was operated in the electron impact mode at 70 eV and scanned from 50 to 650 da. Data were acquired and processed with the Chemstation software (Hewlett-Packard, Palo Alto, CA, USA). Individual compounds were identified by comparison of mass spectra with literature and library data, comparison with authentic standards, and interpretation of GC retention times and mass spectrometric fragmentation patterns. The GC retention times are expressed as Kovats indices (KI) and cited on each respective mass spectrum [79].

## 4. Conclusions

Natural products (diterpenoids) in resins are the best molecular tracers for conifers in the environment and fossil record. Not all compounds survive as such because they are altered or removed by oxidation and diagenetic reactions. However, compounds in resin are protected from oxidation and instead undergo disproportionation or polymerization reactions. The major characteristic markers for the Taxodioideae conifer subfamily are ferruginol, 7α-p-cymenylferruginol and chamaecydin, with secondary salvinolone, *abeo*-carnosol and taxodione acetate. We provided an overview of the natural product precursors of the Taxodioideae to apply in environmental, geological, archeological, forensic and pharmaceutical studies. Furthermore, we summarized the presence of these tracers in the fossil record.

## Figures and Tables

**Figure 1 molecules-24-03036-f001:**
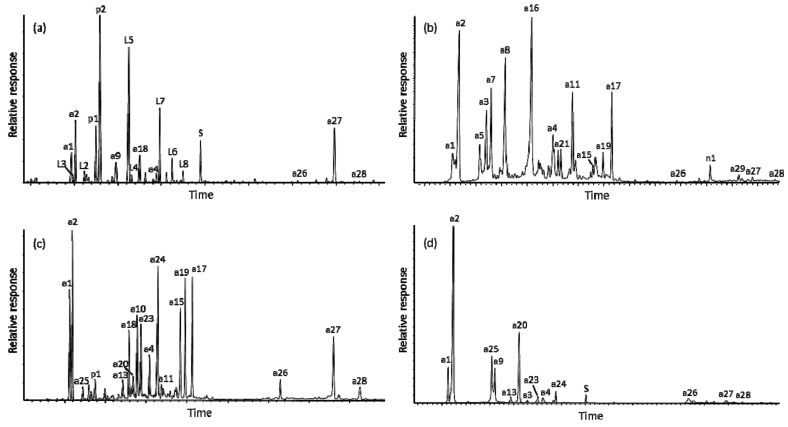
Annotated examples of GC-MS total ion current traces for the total extracts (analyzed as the trimethylsilyl derivatives) of: (**a**) *Cryptomeria japonica* resin from trunk, (**b**) *Glyptostrobus pensilis* resin from cone, (**c**) *Taxodium distichum* resin from trunk, and (**d**) *T. mucronatum* resin from cones. Compound identities are given in Table 2 and structures in Appendix B.

**Figure 2 molecules-24-03036-f002:**
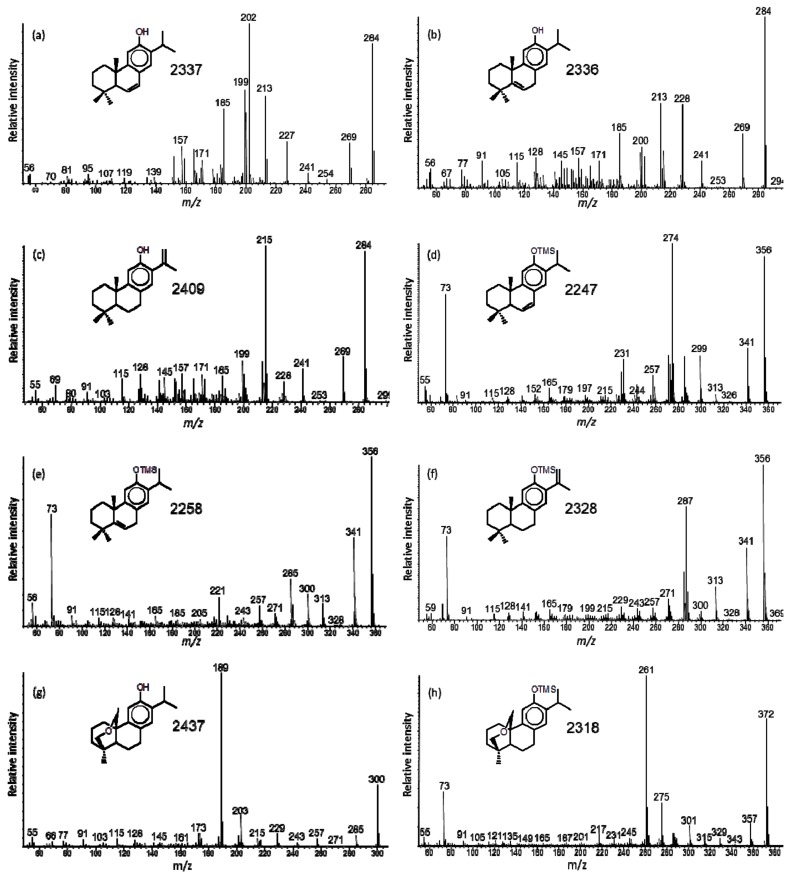
Mass spectra for: (**a**) 6,7-dehydroferruginol (*a1*, extant), (**b**) 5,6-dehydroferruginol (fossil), (**c**) 15,16-dehydroferruginol (fossil), (**d**) 6,7-dehydroferruginol-TMS, (**e**) 5,6-dehydroferruginol-TMS, (**f**) 15,16-dehydroferruginol-TMS, (**g**) 19(20)-oxoferruginol (*a5*), and (**h**) 19(20)-oxoferruginol-TMS.

**Figure 3 molecules-24-03036-f003:**
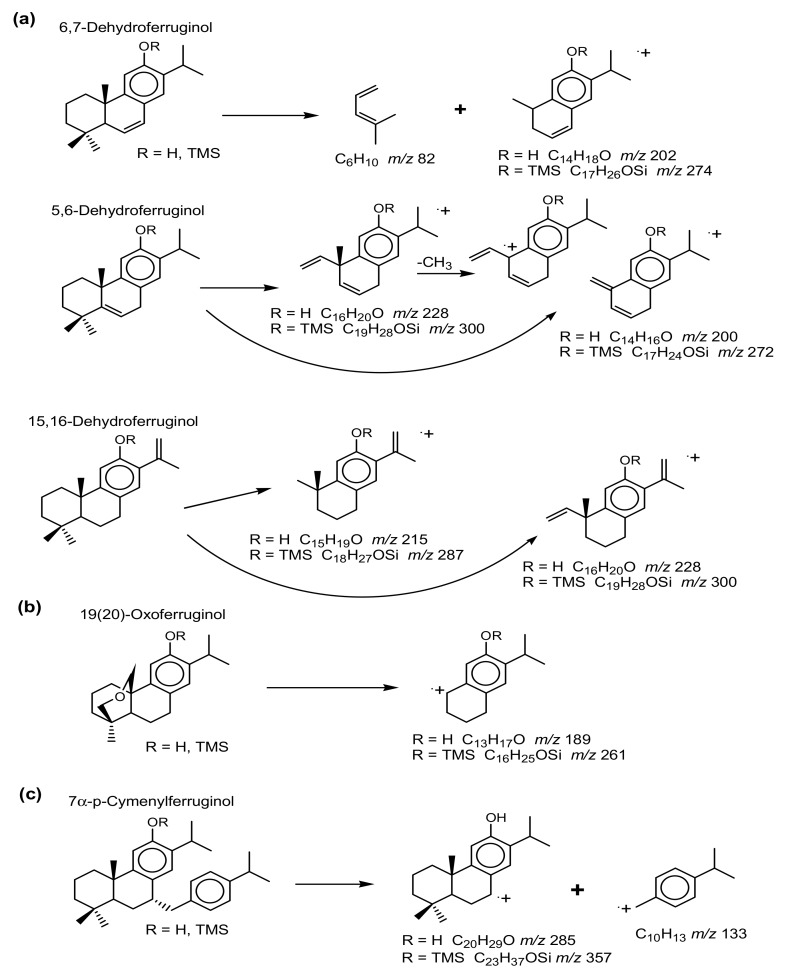
Interpretation of key fragment ions for the mass spectra of Figure 2 and Figure 4.

**Figure 4 molecules-24-03036-f004:**
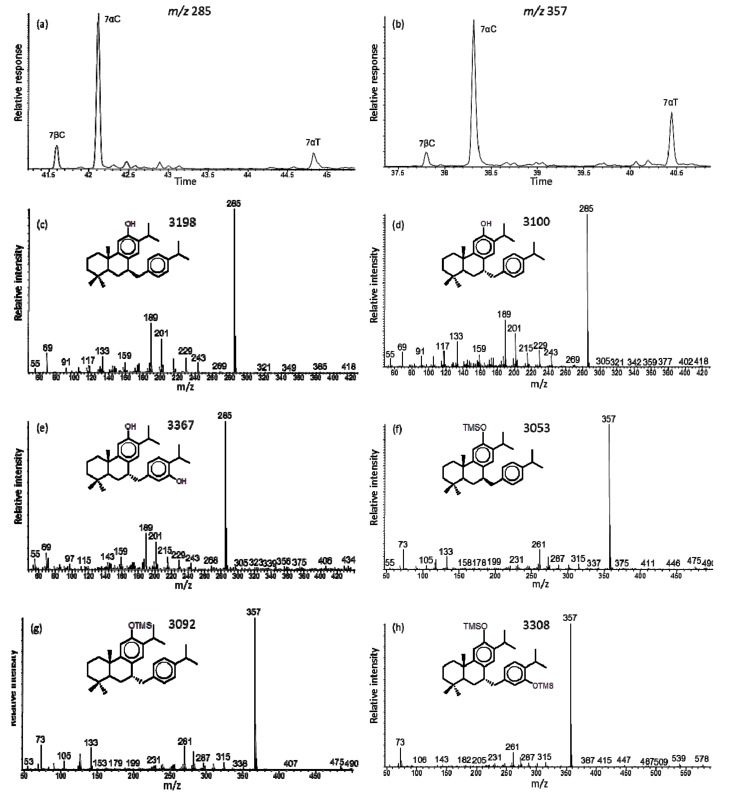
GC-MS data for: (**a**,**b**) Key ion plots of *m/z* 285 and 357 showing the elution of the C_30_ isomers (*a26*), (**c**) mass spectrum of 7β-p-cymenylferruginol (7βC, M ^+^ 418), (**d**) mass spectrum of 7α-p-cymenylferruginol (7αC, M ^+^ 418), (**e**) mass spectrum of 7α-p-thymylferruginol (7αT, M ^+^ 434), (**f**) mass spectrum of 7β-p-cymenylferruginol-TMS (7βC, M ^+^ 490), (**g**) mass spectrum of 7α-p-cymenylferruginol-TMS (7αC, M ^+^ 490), and (**h**) mass spectrum of 7α-p-thymylferruginol-TMS (7αT, M ^+^ 578).

**Table 1 molecules-24-03036-t001:** Sources of plant materials.

Botanical Name	Common Name	Sample Type	Sampling Location	Number of Analyses
*Cryptomeria japonica*	Sugi cedar	Resin on trunk, branchlet	Narita Temple, Tokyo Metro, Japan	7
*Glyptostrobus pensilis*	Chinese swamp cypress	Resin on trunk, cone, shoot	Botanical Institute, Chinese Academy of Sciences, Guangzhou, China	22
		Resin on trunk, cone, shoot	Boise, ID, USA	
*Taxodium distichum*	Swamp cypress	Resin on trunk, shoot	Cypress National Park, FL, USA	22
		Resin on trunk, shoot	Botanical Institute, Chinese Academy of Sciences, Guangzhou, China	
		Resin on trunk, shoot	Lithia Park, Ashland, OR, USA	
*Taxodium mucronatum*	Montezuma cypress	Resin on trunk, cone	Tule Tree, Oaxaca, Mexico	14
		Resin on trunk, cone	Botanical Garden, Los Angeles, CA, USA	

**Table 2 molecules-24-03036-t002:** Diterpenoids in the resins of *Cryptomeria japonica*, *Glyptostrobus pensilis*, *Taxodium distichum*, and *T. mucronatum*.

Number (Appendix B)	Name	Formula ^1^	MW	Relative Abundance ^2^				ID ^3^	References
				***Crytomeria japonica***	***Glyptostr. pensilis***	***Taxodium distichum***	***Taxodium mucronatum***		
	**ALIPHATICS**								
n1	*n*-Nonacosan-10-ol	C_29_H_60_O	424	6	15		8	L	[53]
n2	*n*-Nonacosan-10-one	C_29_H_58_O	422		8			L	[54]
	**SESQUITERPENOIDS**								
s1	Cuparene	C_15_H_22_	202				3	S	[55]
s2	Widdrene (thujopsene)	C_15_H_24_	204				20	S	[55]
s3	Mayurone	C_14_H_20_O	204				5	L	[55]
s4	Widdrol (thujopsol)	C_15_H_24_O	220				4	S	[55]
	**DITERPENOIDS**								
	**Abietanes**								
a1	6,7-Dehydroferruginol	C_20_H_28_O	284	18	20	55	25	L	[31,56]
a2	Ferruginol	C_20_H_30_O	286	40	80	80	**100**	S	[34]
a3	6,7-Dehydropisiferol	C_20_H_28_O_2_	300		48			I	[54]
a4	Sugiol (7-ketoferruginol)	C_20_H_28_O_2_	300	10	30	30	10	S	[54]
a5	19(20)-Oxoferruginol	C_20_H_28_O_2_	300		26			I	
a6	11-Hydroxy-12-oxoabieta-7,9(11),13-triene (or 6-deoxotaxodione)	C_20_H_28_O_2_	300	8		9	21	S,L,I	[55]
a7	Pisiferol	C_20_H_30_O_2_	302		60			L, I	[7,54]
a8	*abeo*-Pisiferol	C_20_H_30_O_2_	302		75			L	[57]
a9	11-Hydroxyferruginol	C_20_H_30_O_2_	302	12			23	L	[58]
a10	Taxodione, R=H	C_20_H_26_O_3_	314			50	10	S,L	[24]
a11	Salvinolone	C_20_H_26_O_3_	314		50	8	1	S	[33]
a12	6,7-Dehydroroyleanone	C_20_H_26_O_3_	314			6	4	S	[34]
a13	Royleanone	C_20_H_28_O_3_	316			10	8	S,L	[24]
a14	Taxodone	C_20_H_28_O_3_	316			**100**	12	S,L	[24]
a15	11-Hydroxysugiol	C_20_H_28_O_3_	316			60		L	[27,31]
a16	*abeo*-Carnosol (demethyl salvicanol)	C_20_H_30_O_3_	318		**100**			L	[59]
a17	6-Hydroxysalvinolone (14-deoxycoleon U)	C_20_H_26_O_4_	330		60	74		S	[33,54]
a18	Taxoquinone	C_20_H_28_O_4_	332	27		40		L	[24]
a19	7-Hydroxytaxodone	C_20_H_28_O_4_	332		24	70		I	[54]
a20	7-*epi*-Taxoquinone (horminone)	C_20_H_28_O_4_	332			12		I	[24]
a21	6α,11-Dihydroxysugiol (5,6-dihydro-6β-hydroxysalvinolone)	C_20_H_28_O_4_	332		35	8		L, I	[31]
a22	6β,11-Dihydroxysugiol (5,6-dihydro-6β-hydroxysalvinolone)	C_20_H_28_O_4_	332		16			I	
a23	6α-Hydroxytaxoquinone	C_20_H_28_O_5_	348			55	6	S,I	[54]
a24	6β-Hydroxytaxoquinone	C_20_H_28_O_5_	348			70	11	S,I	[54]
a10	Taxodione acetate, R=Ac	C_22_H_28_O_4_	356			12	14	I	[54]
a25	7-Acetoxy-6,7-dehydroroyleanone	C_22_H_28_O_5_	372			5	20	S	[54]
a26	7α-p-Cymenylferruginol	C_30_H_42_O	418	2	1	15	5	L	[36]
a27	Chamaecydin	C_30_H_40_O_3_	448	42	1	40	25	S	[26,54]
a28	*Iso*-chamaecydin	C_30_H_40_O_3_	448	2	2	8	4	L	[26,54]
a29	6β-Hydroxychamaecydin	C_30_H_40_O_4_	464	2	3	3	1	S	[26,54]
	**Pimaranes**								
p1	Sandaracopimaric acid	C_20_H_30_O_2_	302	35	18	10		S	
p2	Isopimaric acid	C_20_H_30_O_2_	302	**100**				S	
	**Labdanes**								
L1	Manool	C_20_H_34_O	290	12	5		15	S	
L2	E,Z-Communic acids	C_20_H_30_O_2_	302	11				L	[55]
L3	*Iso*-communic acid	C_20_H_30_O_2_	302	4				L	[60]
L4	Imbricataloic acid	C_20_H_32_O_3_	320	5				S	
L5	13-*epi*-Cupressic acid	C_20_H_32_O_3_	320	75				L	[28,61]
L6	*Iso*-cupressic acid	C_20_H_32_O_3_	320	20				L	[62]
L7	Imbricatoloic acid	C_20_H_34_O_3_	322	48				S	[28,62]
L8	13,14-Dihydroagathic acid	C_20_H_32_O_4_	336	9				S	[54]
	**TRITERPENOIDS**								
T1	24-Methylenecycloartan-3-one	C_31_H_50_O	438		1			I	[54]
T2	24-Ethylenecycloartan-3-one	C_32_H_52_O	452		0.4			I	[54]

^1^ Analyzed as the natural or trimethysilylated compounds in the extract mixtures. ^2^ Relative abundance normalized to major peak = 100%. ^3^ Identification: S = standard; L = published data; I = interpretation of mass spectrometric fragmentation pattern.

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
