# Peer review of "Terpenoids of the Swamp Cypress Subfamily (Taxodioideae), Cupressaceae, an Overview by GC-MS"

_molecules, 2019, doi:10.3390/molecules24173036_

Round 1

Reviewer 1 Report

The manuscript entitled 'Terpenoids of the Swamp Cypress Subfamily
(Taxodioideae), Cupressaceae, an Overview by GC-MS' submitted by Simoneit and coworkers have characterized the resin bleeds from different families by GC-MS which is a nice work done and can be useful for various fields. The manuscript is well presented, however, it needs few corrections.

The first line of abstract has bled instead of bleed, should be corrected.

The title of Table 2 is not completely visible and should be fixed in lansdscape mode.

on page 10, the side product for compound a) 6, 7-Dehydroferruginol, for which m/z is given as 82 is not correct. The manuscript should be checked for other errors. 

Author Response

Thank you for your assessment and comments.

The term bled is correct as far as I know. It means the resin bled in the past and dried, not actively bleeding.

The galley copy I see has full Table 2 title.

Page 10 (a) The m/z 82 fragment is correct as loss of C6H10.

Other errors are fixed.

Reviewer 2 Report

The total compound mixtures in resins from Taxodioideae subfamily is reported. Also data of the GC-MS results and full mass spectra of novel compounds of the resin as free and derivatized products. This information is useful in environmental, geological, archeological, forensic and pharmaceutical studies. The analysis is well done and interpretations of the fractionation of the compounds are included.  

Author Response

Thank you for your assessment

Reviewer 3 Report

Simoneit and colleagues report GC-MS analysis and identification of terpenoids contents of the swamp Cypress subfamily. This article contains a high concentration of useful information which would be useful for future investigators as a reference. I found the design, rationale, supporting data, and conclusion consistent. I recommend publication in Molecules. 

Author Response

Thank you for your assessment